# Potential Effects of Permanent Daylight Savings Time on Daylight Exposure and Risk during Commute Times across United States Cities in 2023–2024 Using a Biomathematical Model of Fatigue

Jaime K. Devine [1,*] , Jake Choynowski [1] and Steven R. Hursh [1,2]

1   Institutes for Behavior Resources, 2104 Maryland Ave, Baltimore, MD 21218, USA;
    jchoynowski@ibrinc.org (J.C.); shursh@ibrinc.org (S.R.H.)
2   Department of Psychiatry and Behavioral Sciences, Johns Hopkins University School of Medicine,
    4940 Eastern Ave., Baltimore, MD 21224, USA
*   Correspondence: jdevine@ibrinc.org

**Abstract:** Background: Permanent Daylight Savings Time (DST) may improve road safety by providing more daylight in the evening but could merely shift risk to morning commutes or increase risk due to fatigue and circadian misalignment. Methods: To identify how potential daylight exposure and fatigue risk could differ between permanent DST versus permanent Standard Time (ST) or current time arrangements (CTA), generic work and school schedules in five United States cities were modeled in SAFTE-FAST biomathematical modeling software. Commute data were categorized by morning (0700–0900) and evening (1600–1800) rush hours. Results: Percent darkness was greater under DST compared with ST for the total waking day (t = 2.59, $p = 0.03$) and sleep periods (t = 2.46, $p = 0.045$). Waketimes occurred before sunrise $63 \pm 41\%$ percent of the time under DST compared with CTA ($42 \pm 37\%$) or ST ($33 \pm 38\%$; $F_{(2,74)} = 76.37$; $p < 0.001$). Percent darkness was greater during morning ($16 \pm 31\%$) and lower during evening rush hour ($0 \pm 0\%$) in DST compared with either CTA (morning: $7 \pm 23\%$; evening: $7 \pm 14\%$) or ST (morning: $7 \pm 23\%$; evening: $7 \pm 15\%$). Discussion: Morning rush hour overlaps with students' commutes and shift workers' reverse commutes, which may increase traffic congestion and risk compared with evening rush hour. Switching to permanent DST may be more disruptive than either switching to ST or keeping CTA without noticeable benefit to fatigue or potential daylight exposure.

**Keywords:** daylight savings time; light exposure; fatigue risk; biomathematical modeling of fatigue; road safety; traffic

## 1. Introduction

Daylight Saving Time (DST) is a period of the year between March and November when clocks in most parts of the United States are set one hour ahead of Standard Time (ST), leading to more sunlight during evening hours. The United States first established DST in 1918, and federal regulations regarding DST have been unchanged since 2007 [1,2]. In recent years, however, 29 states have introduced legislation to abolish the twice-yearly changing of clocks, and in March 2022, the United States Senate passed legislation called the Sunshine Protection Act to make DST permanent starting in 2023 [3].

Proponents of permanent DST argue that more daylight in the evenings would increase physical and economic activity, reduce energy costs, and improve road safety. Opponents of permanent DST argue that shifting the clock permanently forward may result in circadian misalignment and negative health effects as individuals will be forced to start their days before dawn during the winter months [4–6]. The body of research looking into the potential effects of DST on the economy, exercise, or energy costs has produced mixed results [7–12].

While some findings indicate a benefit of permanent DST in these areas, other studies suggest little or even a detrimental effect. Road safety is another area of contention within the debate surrounding permanent DST. Abolishing the twice-yearly transition between standard time and DST has been recommended for improving safety through a reduction in motor vehicle accidents [13]. Some studies argue that permanent DST would have beneficial effects on road safety by shifting more daylight to the evening hours when crash risk is highest [14]. Darkness is a major contributor to motor vehicle accident risk during evening rush hours [13–17]. However, shifting light to the evening hours comes at the cost of light during early morning commutes [15].

A lack of natural sunlight in the morning could not only increase the risk of vehicular accidents during these times but could result in circadian misalignment as individuals are forced to start their day prior to sunrise [4–6]. Circadian misalignment is associated with increased cardiovascular disease risk, metabolic syndrome, and other health risks [4]. Light is the body's strongest zeitgieber, or environmental cue about time. Natural daylight is usually 100 to 1000 times brighter than artificial light, and a lack of exposure to natural sunlight, even with the use of electrical lighting, has been shown to alter circadian physiology and sleep behavior [18]. The time of awakening is additionally correlated with sunrise and tends to be later in the winter [19]. Establishing year-round DST could, therefore, result in population-level sleep disruption and fatigue, particularly during winter months [4,6,18,19].

Increased fatigue due to waking before sunrise is important not only for health reasons but also for road safety. Importantly, if drivers are fatigued, the benefit of better lighting conditions may not translate to a reduction in crash risk. This impact could be especially deleterious for school children. Research shows that delaying school start times benefits students' sleep and daytime function, as well as reducing adolescent motor vehicle crash risk [20–23]. If DST becomes permanent, the benefit of the legislature to delay school start times could essentially be nullified.

Many of the arguments for or against permanent DST hinge on the assumption that individuals' work or school activities start between the hours of 0700 and 0900. These types of schedules would be affected by a one-hour shift in the timing of sunrise. In fact, both proponents of permanent DST and proponents of permanent standard time argue that darkness, either in the mornings or the evenings, respectively, could be avoided by adjusting schedules to avoid activities during these times [5,15]. However, the 16% of the United States population who currently follow shift work schedules [24] would also be affected by changes to sunrise and sunset. Shift workers are at an increased risk of fatigue and sleep problems that may affect their safety and ability to perform [25,26]. While it is known that both DST and shiftwork impact the health and safety of workers [27], the direct impact of time change arrangements on shift workers has not been thoroughly investigated.

If enacted, the Sunshine Protection Act would result in permanent DST in most states beginning in November 2023 on the assumption that the change will benefit society [3]. While the research literature does not definitively support a health or economic advantage to permanent DST, no studies have directly compared the impact of permanent DST against ST or current time arrangements (CTA) in the same location at the same time beforehand. A biomathematical model of fatigue such as the Sleep, Activity, Fatigue, and Task Effectiveness Fatigue Avoidance Scheduling Tool (SAFTE-FAST) can simultaneously predict the impact of permanent DST compared with CTA or permanent ST on fatigue risk and potential daylight exposure across seasons, time zones, and activity schedules in a way that is impossible to achieve using actual real-world data. The current study aims to answer the research question of how time change arrangements differentially affect sleep timing, sleep duration, task effectiveness, and potential daylight exposure between day, evening, and night shift work schedules as well as school schedules and daily commutes in five major United States cities during autumn, winter, spring, and summer conditions during 2023–2024 using the modeling software SAFTE-FAST. The goal of this analysis is to

provide objective, a priori computational data on the impact of time change arrangements for the benefit of transportation safety officials, policymakers, and circadian researchers.

## 2. Materials and Methods

### 2.1. SAFTE-FAST Biomathematical Modeling Software

SAFTE-FAST is a two-step, three-process model that estimates sleep patterns around work duties using a function called AutoSleep and then provides a continuous prediction of Effectiveness as a function of performance on the Psychomotor Vigilance Task (PVT) [28,29]. Effectiveness is expressed as a percentage scaled to a fully rested person's normal best performance on the PVT (e.g., 100%) [29,30]. The higher the score, the lower the fatigue risk. For reference, Effectiveness scores below 77 indicate PVT performance comparable to an individual with equivalent to 18.5 h of continued wakefulness for a fully rested person or a blood alcohol concentration (BAC) of 0.05 g/dL [31,32]. The ability of AutoSleep to predict average sleep behavior (i.e., sleep timing and duration) as a function of work schedules, time of day, and sleep propensity has been successfully evaluated in shift-working operational populations [30,33,34]. AutoSleep predicts sleep as a function of available time outside of work events as well as time of day. As such, evening and night schedules result in a split sleep schedule commonly observed in shift-working populations. SAFTE-FAST solutions are used in transportation and shiftwork environments as part of a fatigue risk management system (FRMS). SAFTE-FAST has previously been used to evaluate accident risk in railroad engineers [35]. Regulators for the Federal Rail Administration (FRA) consider Effectiveness scores at or below 70 to constitute an area of high fatigue risk [36].

The SAFTE-FAST model predicts circadian misalignment by mimicking the process of the internal circadian oscillator as it adjusts to a different time zone or activity schedule. Model-estimated circadian misalignment is incorporated into Effectiveness performance predictions. SAFTE-FAST has been shown to be reliable in predicting circadian misalignment in association with accident analysis in air traffic controllers and railroad crews [37,38]. The current study did not model schedule changes or travel across time zones. Thus, for the purposes of these hypothetical schedules, the only assumed source of circadian misalignment was the bi-annual clock change under CTA conditions.

SAFTE-FAST also has the capability to model a buffer around work events to indicate the time during which individuals would reasonably be expected to be commuting to or from a work location. Furthermore, SAFTE-FAST contains a NASA-provided algorithm for determining the available sunlight for any location on the globe for any date and time. SAFTE-FAST uses this light information to indicate the degree of concordance between the sleep-wake pattern and the rising and setting of the sun and, by implication, determine the phase shift effects associated with the onset and offset of DST time changes or to extrapolate information about potential daylight exposure during sleep, commute times, working hours, and across the entire day. While this information is not a standard output in the software, the software parameters were adapted to allow the extraction of light data in addition to predicted performances for the analyses described herein. The model cannot account for individual differences, such as time spent indoors or chronotype at this time, but it can provide an estimate of the potential amount of daylight to which individuals could be exposed under different conditions. Additionally, the model can be used to estimate lighting conditions on the road during model-identified commute times.

### 2.2. City Selection Criteria

#### 2.2.1. Location and Observation of DST

SAFTE-FAST predicts sunrise and sunset as a function of location and date. Locations were selected based on their ability to model the range of effects that time change arrangements may have on light and risk across regions that would be affected by the Sunshine Protection Act [3], i.e., states within the United States (U.S.) that currently use temporary DST rather than permanent ST. U.S. states Arizona and Hawaii and the U.S. territories of

American Samoa, Guam, the Northern Mariana Islands, Puerto Rico, and the Virgin Islands observe permanent ST and are not eligible for inclusion [39].

Calculating sunrise and sunset for any given day and year requires latitude and longitude in degrees as input [40]. Day length is fairly consistent across the seasons in latitudes close to the Equator (~0–23°) and varies to an extreme degree in the Artic circle (~66–90°). In order to select locations that would experience seasonal variation, locations needed to be between 23° N and 66° N, as well as at least 2° different from the other selected locations.

There are five time zones within the U.S. that observe DST—Eastern Time, Central Time, Mountain Time, Pacific Time, and Alaska Time [5,41]. While these time zones are not strictly determined by longitude, the U.S. time zones are defined in the Uniform Time Act roughly by degree of longitude west from Greenwich [42], as shown in Figure 1. In order to model a range of U.S. cities, each location needed to be in a different U.S. time zone.

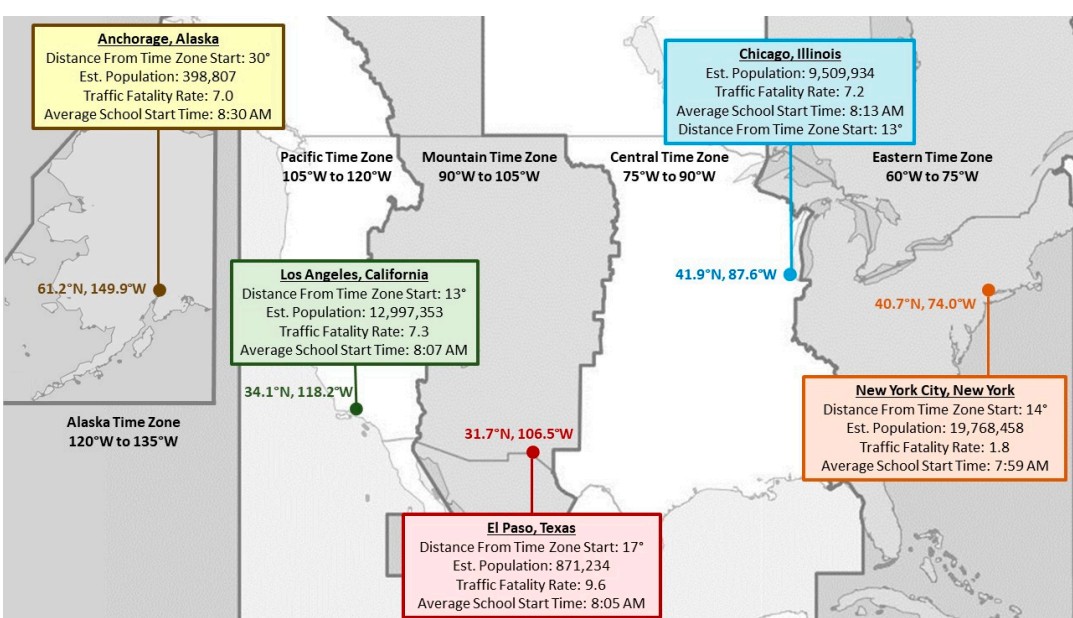

**Figure 1.** City Locations Selected for Modeling. Map of the five metropolitan statistical areas selected for modeling based on time zone by longitude range (in black), with city and state names, latitude, longitude, distance of the city in degrees west from the start of the time zone, estimated population, and traffic fatality rates (listed in box). The average school start times for each selected city's state are additionally listed in the boxes.

### 2.2.2. Population and Traffic Congestion

Since the goal of this computational analysis was to evaluate risk in relation to road safety, selection criteria included risk due to traffic based on highway fatality rates and population size. Inclusion criteria for cities required a highway fatality rate for the county that was greater than 1.0 per 100,000 inhabitants as reported by the National Highway Traffic Safety Administration's (NHTSA) State Traffic Safety Information (STSI) report [43]. Locations also needed to meet the U.S. Census Bureau criteria for metropolitan statistical areas by having at least one urbanized area of 50,000 or more inhabitants [44]. The largest metropolitan statistical area in a given time zone that met all inclusion criteria was selected for subsequent analysis. The five selected city locations were (1) New York City, New York; (2) Chicago, Illinois; (3) El Paso, Texas; (4) Los Angeles, California; and (5) Anchorage, Alaska.

### 2.3. Selection of Time Periods

Four 30-day time periods were selected for modeling based on seasonal variation in day length based on solstices and DST changeover. Dates were selected to represent time

periods after the potential enactment of the Sunshine Protection Act [3] in November 2023. The winter solstice is scheduled to occur on 21 December 2023, and the summer solstice is scheduled to occur on 20 June 2024. DST ends on the first Sunday in November and starts on the second Sunday in March [3,42]. This corresponds to 5 November 2023 and 10 March 2024. Autumn schedules were generated for 1–30 November 2023; winter schedules were generated from 15 December 2023 to 15 January 2024; spring schedules were generated for 1–31 March 2024; and summer schedules were generated for 15 June–15 July 2024.

*2.4. Generation of Work Schedule Data*

Three work schedules were selected for modeling on the basis of start and end times relative to sunrise and sunset under different time change arrangements—a typical day shift (0900–1700), an evening shift beginning around sunset (1700–0100), and an overnight shift ending around sunrise (2300–0700). These schedules correlate with the national distribution of work schedules between day (84% of workers), evening (6% of workers), and night shifts (4% of workers), as reported by the United States Bureau of Labor Statistics [24]. Each schedule included a 40 h work week with 8 h shifts occurring during weekdays. An exception was that overnight shifts began at 2300 on Sundays in order to end on Monday mornings. A month's worth of schedules for each shift was generated in order to produce a monthly average prediction of Effectiveness and potential daylight exposure.

*2.5. Generation of School Schedule Data*

School schedules for each location were based on average start times and hours of school per day by state using data from the U.S. Department of Education, National Center for Education Statistics, Schools and Staffing Survey (SASS) [45,46]. A generic school schedule was generated for each selected location for autumn, winter, and spring, as described in Section 2.3. Summer schedules were not modeled since most schools are out of session during this time. For the purposes of this model, the school was assumed to be out of session for Thanksgiving break (23–24 November 2023), winter break (25 December 2023–1 January 2024), and spring break (25–29 March 2024).

*2.6. Modeling Time Change Arrangements in SAFTE-FAST*

Documentation detailing the initial development and validation of the SAFTE model, including model equations and system diagrams, are described in Hursh 2003 and Hursh et al., 2004 [29,47]. A modified version of SAFTE-FAST software (Version 5.8.028) that reports potential daylight exposure by event was used to model work and school schedules by season, city, and time change condition. A separate SAFTE-FAST project file was created for each time change condition (CTA, Permanent DST, and Permanent ST). Identical work and school schedule data were uploaded into each scenario. The only differences between scenarios were time change conditions. AutoSleep is a sleep estimator in SAFTE-FAST that uses information about work events, time of day, and prior sleep to predict average sleep decisions under operational constraints. Documentation regarding the development and validation of the AutoSleep algorithm was published by Roma et al., 2012, Devine et al., 2022, Gertler et al., 2012, and Hursh et al., 2011 [30,33,48,49]. AutoSleep predicted sleep episodes around school or work events using default settings. No sleep was assumed to occur during work hours for these analyses, and no napping was assumed to occur. A 60 min commute time buffer was assumed for the hour before work/school starts and the hour after the conclusion of work/school. Events were categorized as sleep, wake, work, or commute using the SAFTE-FAST Activity and Description output columns. Cognitive performance in SAFTE-FAST is predicted using a metric called Effectiveness. Effectiveness scores are based on reaction time speed on the Psychomotor Vigilance Task (PVT). Effectiveness is expressed as a percentage of individual optimum performance (e.g., 100); lower Effectiveness scores indicate slower cognitive reaction times. Further documentation about how Effectiveness is computed in SAFTE-FAST can be found in Hursh et al., 2004 [29]. Event timing, average and minimum Effectiveness, and potential

daylight exposure (measured as minutes of daylight, minutes of twilight, and minutes of darkness) were exported to .csv files.

*2.7. Statistical Analysis*

SAFTE-FAST CSV files were compiled in Excel 2013. Average and minimum Effectiveness scores were averaged across all days within each season (autumn, winter, spring, and summer) to create a seasonal mean prediction of Effectiveness for each schedule and condition. Average AutoSleep duration and waketime provided predicted values of expected sleep duration and morning waketime for each schedule.

Potential daylight exposure was estimated as the percent of darkness (minutes of darkness/total minutes of event * 100) for sleep, work, commute, and overall wake events. For the purposes of these analyses, minutes of twilight and minutes of daylight were aggregated to reflect time periods with any amount of ambient daylight. Prediction of sunrise and sunset times for 2023–2024 were extracted from the sunearthtools.com Sunrise Sunset Calculator [50]. Distance between AutoSleep expected morning waketimes and sunrise, in minutes, were computed for each modeled day by subtracting sunrise from waketime and averaged across all days by season for each schedule and condition. Positive values indicate waketimes occurring after sunrise, while negative values indicate wake times occurring before sunrise. AutoSleep events were assigned a binary distinction to indicate waketimes occurring before sunrise (1) versus waketimes occurring after sunrise (0). The percentage of waketimes occurring before sunrise for each condition was computed as the total number of waketimes occurring before sunrise over the total number of waketimes for all major periods of sleep by season for each schedule and condition.

Schedules were identified using numeric codes containing information about location (New York City, Chicago, El Paso, Los Angeles, or Anchorage), shift type (day, evening, night, or school), and season (autumn, winter, spring, summer). Repeated measures of analysis of variance (ANOVA) were used to compare differences between time change conditions (CTA, Permanent DST, and Permanent ST) for sunrise time, waketime, and expected sleep duration, controlling for schedule specifics (location, shift, and season). The ANOVA *F* test is generally robust to violations of variance when sample sizes are equal [51]. Repeated measures of ANOVA were further used to compare differences between time change conditions for average and minimum Effectiveness and percent darkness for the total waking day, commute-to-work, workday, and commute-home.

Commute times differed by schedule and did not reflect Effectiveness during rush hours for all schedules. To estimate the risk associated with morning and evening rush hours, Effectiveness and percent darkness during commute times were additionally categorized by time of commute. Morning rush hour was defined as any commute times occurring between 0700–0900, and evening rush hour was defined as any commute times occurring between 1600–1800. Morning rush hour Effectiveness and percent darkness thereby corresponded to commute-to-work values for day and school schedules and commute-home values for night schedules. Evening rush hour Effectiveness and percent darkness corresponded to commute-home values for day schedules and commute-to-work values for evening schedules. Differences in average and minimum Effectiveness and percent darkness for morning and evening rush hours were compared using repeated measures in time ANOVA, controlling for schedule. Time change conditions were treated as a repeated measure for all evaluated variables. Assumptions of normality of distribution for all variables were tested using the skewness/kurtosis tests for normality. The independence of observations can be assumed due to the computational nature of these generated data. Effect sizes were computed as eta-squared ($\eta^2$) using the estat esize function in STATA. Effect sizes are interpreted as $\eta^2 \geq 0.01$, indicating a small effect, $\eta^2 \geq 0.06$, indicating a medium effect, and $\eta^2 \geq 0.14$, indicating a large effect, based on accepted rules of thumb. All statistical analyses were performed in Stata MP 15.

## 3. Results

### 3.1. Schedule Descriptive Statistics

Figure 1 depicts the selected locations by time zone, population, traffic fatality rate, and average school start time. Three separate time change condition scenarios were constructed in SAFTE-FAST to model four different work schedules across five cities during four seasons for work schedules or three seasons for school schedules for a total of 75 schedules per scenario and 225 total schedules. Time change conditions (CTA, permanent DST, and permanent ST) were the only differences between SAFTE-FAST scenarios. Shift start time, shift end time, expected morning waketime, and expected sleep duration did not differ between scenarios (all $p > 0.2$) and are summarized in Table 1. AutoSleep predicts split sleep schedules for evening and night workers; the first waketime occurring during any modeled day served as the expected morning waketime in these subgroups. School start times vary by state, as depicted in Figure 1.

**Table 1.** Waketimes and Sleep Duration by Schedule.

| Shift Type | Shift Start Time | Shift End Time | Expected Morning Waketime | Average Expected Sleep Duration per 24 h (in mins) |
|---|---|---|---|---|
| Day | 09:00 | 17:00 | 07:16 ± 00:04 | 482 ± 45 |
| Evening | 17:00 | 01:00 | 07:25 ± 00:04 | 363 ± 60 |
| Night | 23:00 | 07:00 | 07:53 ± 00:06 | 338 ± 43 |
| School | 08:10 ± 00:17 | 14:45 ± 00:27 | 07:14 ± 00:04 | 460 ± 67 |

### 3.2. Time Change Arrangements and Exposure to Daylight

Figure 2 depicts differences in hours of daylight by season, shift, and city location between time change conditions. There were expected main effects of city location, shift type, and season on exposure to daylight; these results are included in Supplementary Data Table S1. Table 2 summarizes the repeated measures of ANOVA results for exposure to daylight by time change conditions (ST, DST, CTA), controlling for city location, shift type, and season. Skewness and kurtosis values for sunrise, the distance between sunrise and work, and all percent darkness variables described in Table 3 were near zero, indicating normal distribution, symmetrical skewness, and platykurtic kurtosis. The number of observations for each of these analyses was 225. Bonferroni's post hoc analysis revealed significant differences between all conditions for average sunrise, the distance between sunrise and waketime, and the percentage of waketimes occurring before sunrise (all $p \leq 0.001$). There were significant differences between DST and ST conditions for percent darkness during the total waking day ($t = 2.59$, $p = 0.03$) and percent darkness during sleep ($t = 2.46$, $p = 0.045$), but not between either DST or ST and CTA (all $p > 0.3$). There were significant differences between DST and either CTA ($t = 3.05$, $p = 0.008$) or ST ($t = 3.19$, $p = 0.005$) for percent darkness during the commute-to-work, but not between CTA and ST ($t = 0.14$, $p = 1.0$). There were no significant differences between the percent darkness during the work day or commute-home between conditions (all $p > 0.6$).

### 3.3. Time Change Arrangements and Predicted Effectiveness

Table 3 summarizes differences in Effectiveness scores between time change conditions. Skewness and kurtosis values for all Effectiveness variables described in Table 3 were near zero, indicating normal distribution, symmetrical skewness, and platykurtic kurtosis. Repeated measures of ANOVA were performed to compare the effect of time change conditions (ST, DST, CTA) on average and minimum predicted Effectiveness across the year, controlling for city location, shift type, and season. Repeated measures of ANOVA with this model were conducted separately for commute-to-work, work day, commute-home, and total waking day predictions of Effectiveness. The number of observations for each of these analyses was 225. Bonferroni's post hoc analysis revealed that average and minimum Effectiveness scores were significantly higher under either permanent DST or ST conditions

compared with CTA for commute-to-work, work day, commute-home, and total waking day (see Table 3; all $p \leq 0.001$). There were no significant differences between DST and ST (all $p > 0.9$). There were no differences between time change conditions for minimum or average Effectiveness during the total waking day, controlling for city location, shift type, and season (all $p > 0.9$). There were expected differences in Effectiveness by shift and season. Effectiveness did not differ by city location. These results are summarized in Supplementary Data Table S2. Differences in Effectiveness during CTA were driven by dips during March and November related to clock changing (see Supplementary Data Table S2).

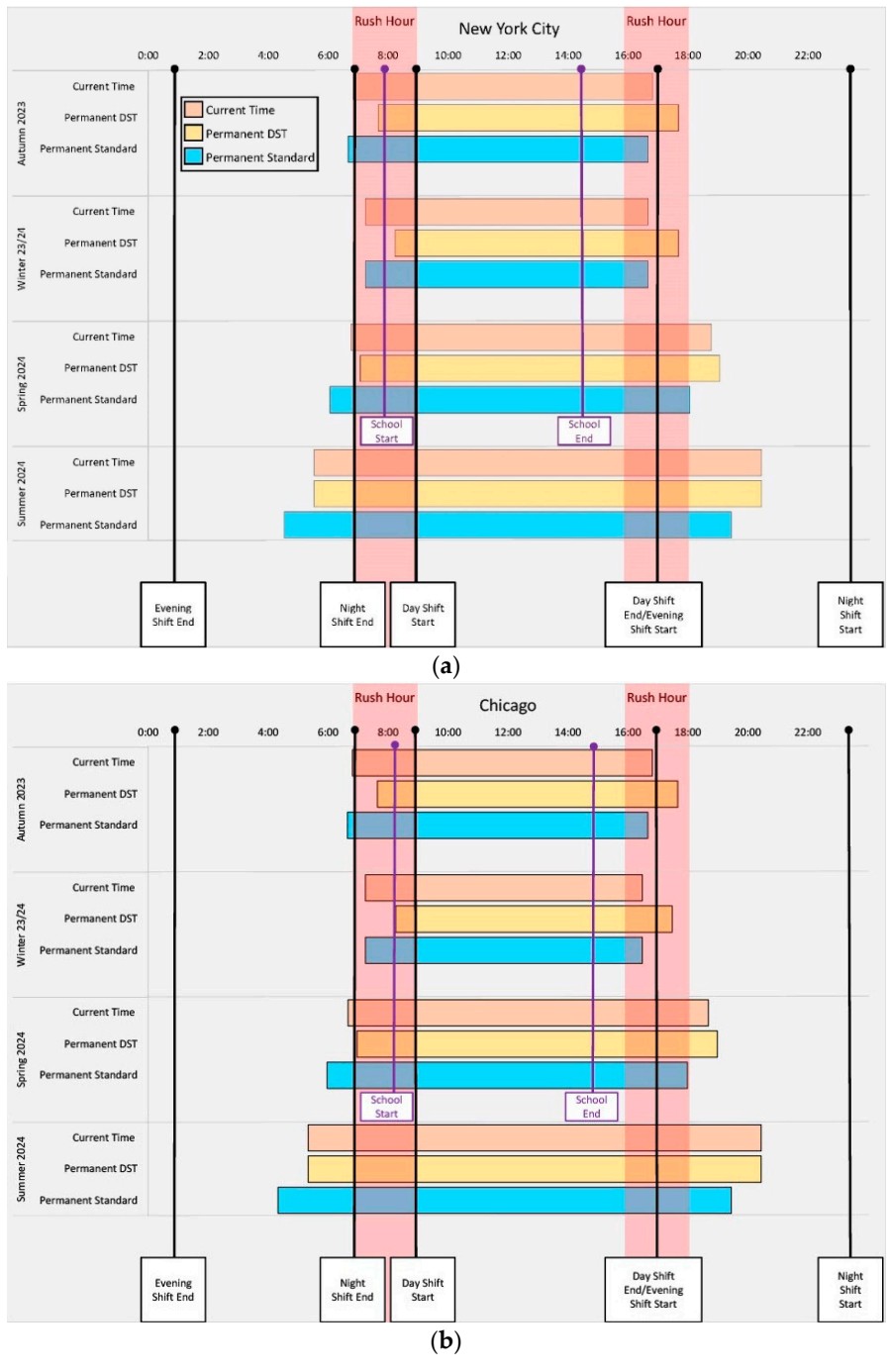

**Figure 2.** *Cont.*

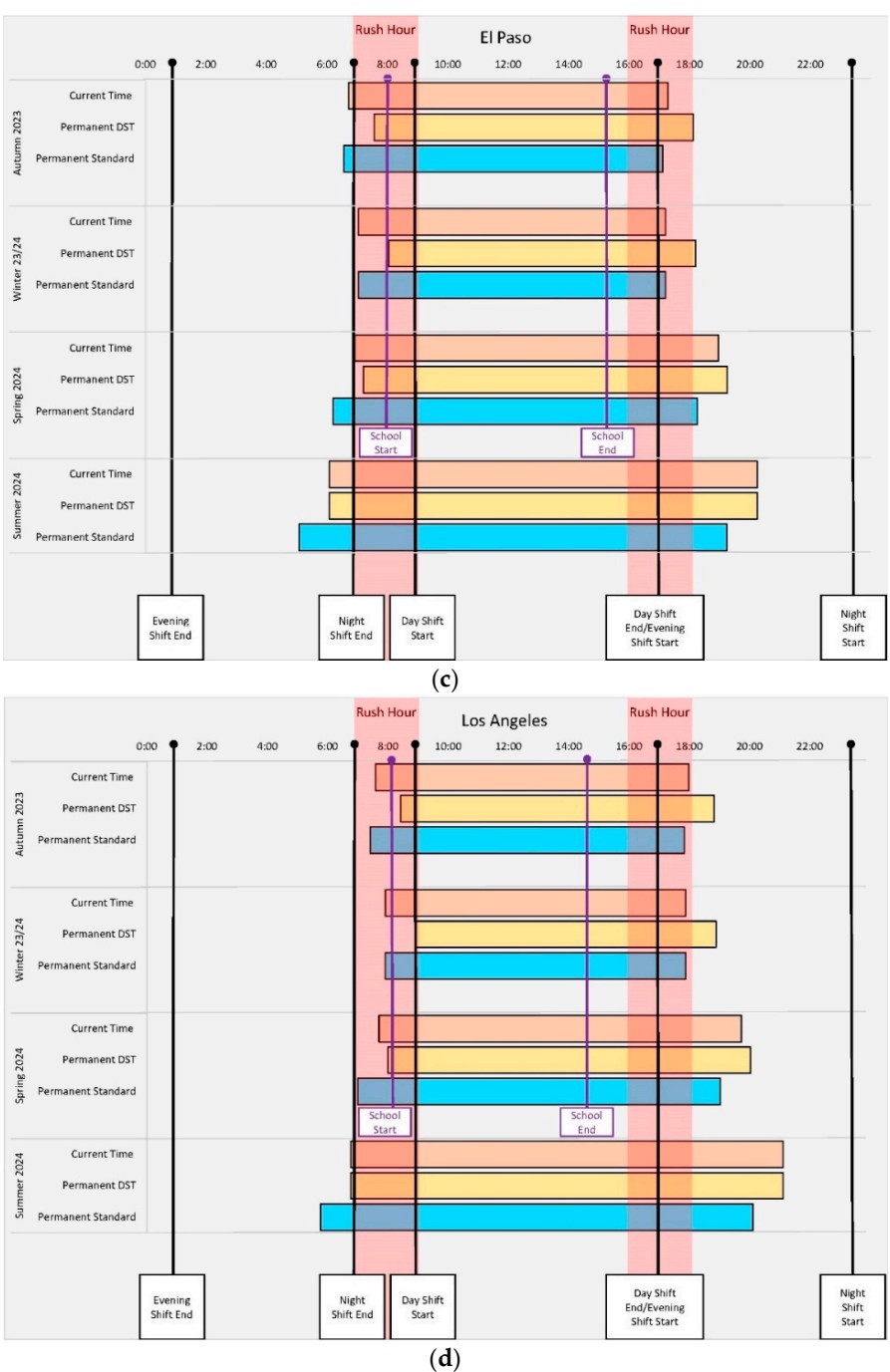

**Figure 2.** *Cont*.

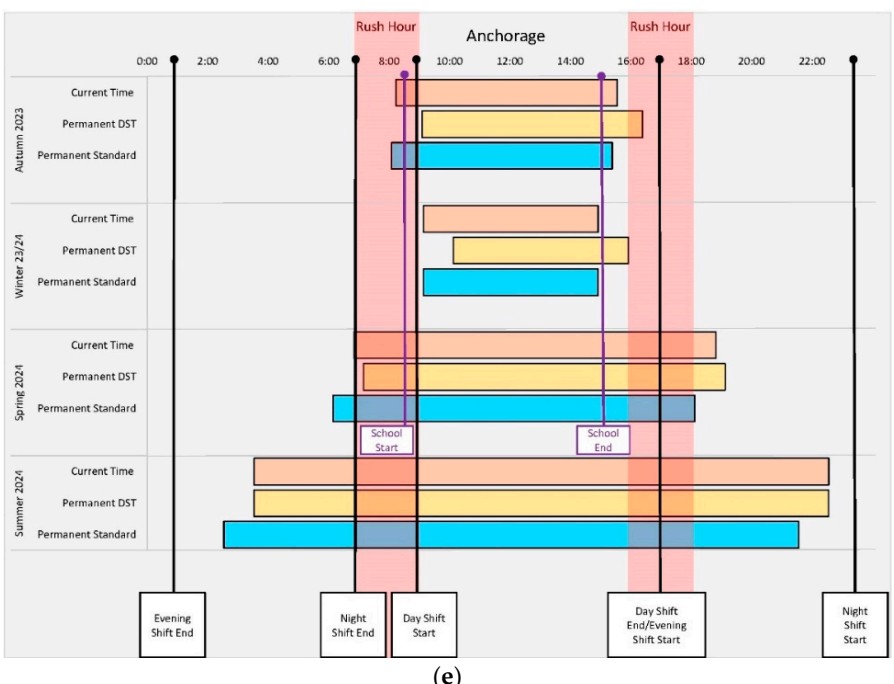

(**e**)

**Figure 2.** Exposure to Daylight by Schedule and Time Change Conditions. Graphic depiction of hours of daylight across the 24-h day (top x-axis) by time change condition (CTA: orange bar, DST: yellow bar, ST: blue bar) and season on the y-axis for (**a**) New York City, (**b**) Chicago, (**c**) El Paso, (**d**) Los Angeles, and (**e**) Anchorage. Work schedule start and end times are indicated by black lines. School start and end times are indicated by purple lines. Morning and evening rush hours are indicated by red shading.

**Table 2.** Effects of Time Change Conditions on Exposure to Daylight.

| | CTA (M ± SD) | Permanent DST (M ± SD) | Permanent ST (M ± SD) | $F_{(2,74)}$ Value | *p* Value | $\eta^2$ (95% CI) |
|---|---|---|---|---|---|---|
| Average Sunrise | 07:08 ± 02:48 | 07:35 ± 03:18 | 06:42 ± 03:18 | 387.24 | <0.001 ** | 0.84 (0.79–0.87) |
| Distance between Sunrise and Waketime ‡ | 19 ± 69 min | −15 ± 86 min | 44 ± 86 min | 406.82 | <0.001 ** | 0.85 (0.80–0.87) |
| Percentage of Waketimes Occurring Before Sunrise | 42 ± 37% | 63 ± 41% | 33 ± 38% | 76.37 | <0.001 ** | 0.51 (0.39–0.59) |
| Percent Darkness During the Total Waking Day | 32 ± 13% | 29 ± 13% | 33 ± 12% | 94.69 | <0.001 ** | 0.56 (0.45–0.64) |
| Percent Darkness During Commute-to-work | 28 ± 44% | 32 ± 45% | 28 ± 44% | 6.48 | 0.002 * | 0.08 (0.01–0.17) |
| Percent Darkness During Work Day | 41 ± 42% | 41 ± 42% | 41 ± 42% | 0.08 | 0.93 | 0.001 (0.00–0.01) |
| Percent Darkness During Commute-Home | 31 ± 44% | 30 ± 45% | 31 ± 44% | 0.37 | 0.68 | 0.005 (0.00–0.04) |
| Percent Darkness During Sleep | 66 ± 24% | 71 ± 26% | 63 ± 24% | 103.11 | <0.001 ** | 0.58 (0.48–0.65) |

‡ Negative values indicate waketimes occurring before sunrise. Positive values indicate waketimes occurring after sunrise. * indicates *p* values ≤ 0.05; ** indicates *p* values ≤ 0.001. $\eta^2 \geq 0.01$ indicates a small effect; $\eta^2 \geq 0.06$ indicates a medium effect; $\eta^2 \geq 0.14$ indicates a large effect.

### 3.4. Time Change Arrangements and Rush Hour Commutes

Table 4 summarizes differences in Effectiveness scores and percent darkness during morning rush hour (0700–0900) and evening rush hour (1600–1800) between time change conditions. Morning rush hour included commute-to-work Effectiveness and percent darkness values for day and school shift schedules and commute-home Effectiveness and percent darkness values for night shift schedules for a total of 55 schedules. Repeated measures of ANOVA examined the difference between time change conditions (ST, DST, CTA)

for morning rush hour Effectiveness and percent darkness, controlling for city location, shift type, and season. The number of observations for each of the morning rush hour analyses was 165. There was a trend for differences in average and minimum Effectiveness scores by time change conditions during morning rush hours. Bonferroni's post hoc analysis revealed lower minimum and average effectiveness under CTA compared with either DST or ST (all $p \leq 0.002$). There were significant differences in percent darkness during morning rush hour. Bonferroni's post hoc analysis indicated that there was a greater percentage of darkness under DST conditions compared with either CTA ($t = 4.52$, $p < 0.001$) or permanent ST ($t = 4.75$, $p < 0.001$). There were no differences in percent darkness between CTA and ST ($t = 0.23$, $p > 1.00$).

**Table 3.** Effects of Time Change Conditions on Average and Minimum Effectiveness.

| | CTA (M ± SD) | Permanent DST (M ± SD) | Permanent ST (M ± SD) | $F_{(2,74)}$ Value | $p$ Value | $\eta^2$ (95% CI) |
|---|---|---|---|---|---|---|
| Commute-to-work Average Effectiveness | 96.02 ± 3.27 | 96.12 ± 3.12 | 96.12 ± 3.14 | 9.50 | 0.001 ** | 0.11 (0.03–0.21) |
| Commute-to-work Minimum Effectiveness | 94.94 ± 3.87 | 95.04 ± 3.73 | 95.04 ± 3.75 | 8.24 | 0.004 * | 0.10 (0.02–0.19) |
| Work Day Average Effectiveness | 91.21 ± 10.76 | 91.32 ± 10.60 | 91.33 ± 10.61 | 11.40 | <0.001 ** | 0.13 (0.04–0.23) |
| Work Day Minimum Effectiveness | 86.70 ± 12.32 | 86.84 ± 12.13 | 86.83 ± 12.13 | 4.82 | 0.009 * | 0.06 (0.004–0.14) |
| Commute-Home Average Effectiveness | 87.36 ± 11.83 | 87.50 ± 11.64 | 87.49 ± 11.63 | 3.37 | 0.04 * | 0.04 (0.00–0.12) |
| Commute-Home Minimum Effectiveness | 86.31 ± 11.93 | 86.45 ± 11.75 | 86.44 ± 11.75 | 3.23 | 0.04 * | 0.04 (0.00–0.11) |
| Total Waking Day Average Effectiveness | 93.60 ± 5.99 | 93.63 ± 5.98 | 93.63 ± 5.98 | 9.19 | <0.001 ** | 0.11 (0.03–0.20) |
| Total Waking Day Minimum Effectiveness | 89.45 ± 5.91 | 89.48 ± 5.90 | 89.48 ± 5.89 | 24.26 | <0.001 ** | 0.25 (0.13–0.35) |

\* indicates $p$ values $\leq 0.05$; ** indicates $p$ values $\leq 0.001$. $\eta^2 \geq 0.01$ indicates a small effect; $\eta^2 \geq 0.06$ indicates a medium effect; $\eta^2 \geq 0.14$ indicates a large effect.

**Table 4.** Effects of Time Change Conditions on Rush Hour Effectiveness and Exposure to Light.

| | CTA (M ± SD) | Permanent DST (M ± SD) | Permanent ST (M ± SD) | F Value | $p$ Value | $\eta^2$ (95% CI) |
|---|---|---|---|---|---|---|
| Morning Rush Hour Average Effectiveness | 87.89 ± 14.37 | 88.05 ± 14.15 | 88.07 ± 14.16 | $F_{(2,54)} = 3.14$ | 0.05 † | 0.05 (0.00–0.14) |
| Morning Rush Hour Minimum Effectiveness | 86.92 ± 14.22 | 87.06 ± 14.00 | 87.09 ± 14.00 | $F_{(2,54)} = 2.90$ | 0.06 † | 0.05 (0.00–0.14) |
| Percent Darkness During Morning Rush Hour | 7 ± 23% | 16 ± 31% | 7 ± 23% | $F_{(2,54)} = 14.35$ | <0.001 ** | 0.21 (0.08–0.33) |
| Evening Rush Hour Average Effectiveness | 97.48 ± 0.92 | 97.50 ± 0.90 | 97.49 ± 0.91 | $F_{(2,39)} = 0.54$ | 0.58 | 0.01 (0.00–0.08) |
| Evening Rush Hour Minimum Effectiveness | 97.11 ± 0.68 | 97.12 ± 0.66 | 97.12 ± 0.67 | $F_{(2,39)} = 0.51$ | 0.60 | 0.01 (0.00–0.08) |
| Percent Darkness During Evening Rush Hour | 7 ± 14% | 0 ± 0% | 7 ± 15% | $F_{(2,39)} = 8.80$ | <0.001 ** | 0.18 (0.33–0.62) |

† indicates $p$ values $\leq 0.1$; ** indicates $p$ values $\leq 0.001$. $\eta^2 \geq 0.01$ indicates a small effect; $\eta^2 \geq 0.06$ indicates a medium effect; $\eta^2 \geq 0.14$ indicates a large effect.

Evening rush hour included commute-home Effectiveness and percent darkness values for day shift schedules and commute-to-work Effectiveness and percent darkness values for evening shift schedules for a total of 40 schedules. Repeated measures of ANOVA examined the difference between time change conditions (ST, DST, CTA) for evening rush hour Effectiveness and percent darkness, controlling for city location, shift type, and season. The number of observations for each of the evening rush hour analyses was 120. There were no significant differences in average and minimum Effectiveness scores by time change conditions during evening rush hours. There were significant differences in percent darkness during evening rush hours. Bonferroni's post hoc analysis indicated that there was a greater percentage of darkness under either CTA ($t = 5.04$, $p < 0.001$) or permanent ST

(t = 5.23, *p* < 0.001) compared with permanent DST. There were no significant differences in percent darkness between CTA and ST (t = 0.00, *p* = 1.00). Skewness and kurtosis values for all variables described in Table 4 were between zero and one, indicating normal distribution, symmetrical skewness, and platykurtic kurtosis. Breakdowns of average Effectiveness and percent darkness during morning and evening rush hours by city, season, and time change condition, with information about included schedules, are included in Supplementary Data Tables S3 and S4, respectively.

## 4. Discussion

The purpose of this computational modeling project has been to evaluate the average potential impact that time change arrangements alone may have on cognitive alertness and exposure to daylight in United States locations under a variety of seasons and work or school schedules. To our knowledge, this is the first attempt to model the impact of time change arrangements using a biomathematical model of fatigue (SAFTE-FAST) with a sleep prediction algorithm (AutoSleep). Our findings suggest that under ideal hypothetical circumstances, abandoning the twice-yearly clock change may be nominally beneficial for Effectiveness. Permanent DST conditions resulted in less light at waketime, during morning rush hour, and less potential daylight exposure across the day than either CTA or ST. Given the similarities between CTA and ST in these analyses, it appears that adjusting to permanent ST may be logistically easier than adapting to permanent DST time conditions.

With regards to Effectiveness, these simulated data suggest that adopting either permanent DST or permanent ST may prevent cognitive alertness deficits related to the bi-annual transition between ST and DST in November and March (see Table 3). Although they are statistically significant, the observed differences in predicted Effectiveness are less than a full integer, and the effect sizes indicate only a medium effect. Moreover, scores are above the FRA cut-off for fatigue risk (an Effectiveness score of 70) [36]. Time change arrangements did not show a significant effect on rush hour Effectiveness in this analysis (see Table 4) either. Taken together, it is unlikely that fatigue risk would be noticeably different based on the time change conditions alone. Previous research investigating the contributing role of DST transitions on cognitive performance or accident risk has shown mixed results, with some studies indicating an increased risk due to clock changing and other studies showing no association [13,51–53]. The risk of fatigue due solely to the bi-annual clock change may be negligible under ideal conditions, such as fixed schedules that consistently allow for a sufficient amount of sleep, but could interact with other factors to produce higher risk in real-life situations.

Deficits in alertness due to the clock change may reasonably be compounded by individual differences in sleep behavior, work schedules, or resilience to fatigue that could be variable across populations. These differences could potentially account for the mixed findings with respect to the impact of time changes on accident risk seen in real-world data analyses. The AutoSleep algorithm predicts sleep as a function of time available between work events and will assume an 8 h, overnight sleep opportunity unless time is constrained by the work schedule. The schedules modeled in this analysis may be considered representative of ideal sleep and working conditions. Individual differences in sleep behavior or cognitive alertness, including behavior related to time change arrangements rather than potential daylight exposure, cannot be predicted using generic fixed schedules and the AutoSleep function in SAFTE-FAST. Furthermore, AutoSleep has not been evaluated for sleep prediction in student populations. The use of a sleep prediction algorithm rather than actual measures of sleep behavior under different time change conditions constitutes a limitation for the interpretability of the presented results. It is possible to model objective measures of sleep in SAFTE-FAST to produce a more specific prediction of Effectiveness. However, since it is not possible to collect ecologically valid sleep data across seasons in the future (years 2023–2024) in multiple cities simultaneously under three different time change conditions, AutoSleep provides an adequate exploratory proxy for real-world sleep in this analysis.

Setting the clocks forward in the spring has been shown to disrupt sleep and impair cognitive performance, as well as shift the amount of light available during morning commutes compared with evening commutes [13,14,17,51–53]. Decreasing the amount of darkness during evening rush hour to reduce crash risk is an argument for the adoption of permanent DST [14,17]. As expected, the percent darkness during morning rush hour in this study was greater under permanent DST conditions compared with CTA or permanent ST (16% vs. 7%; see Table 4), while the percent darkness during evening rush hour was lower under permanent DST conditions compared with the other conditions (0% vs. 7%; see Table 4). However, Effectiveness during morning rush hour was lower than Effectiveness during evening rush hour (see Table 4). This difference can be attributed to the inclusion of shiftwork schedule commute data. Morning rush hour coincided with commute-home data from night schedules when workers are assumed to have lower Effectiveness following a full 8 h of work, whereas evening rush hour included commute-to-work data from evening schedules, when workers are assumed to be well-rested.

Shift workers are rarely considered in the discussion of the impact of time change arrangements on highway safety. Our simulated dataset suggests that more individuals would be on the road during morning rush hour than evening rush and that fatigue risk would be greater due to shift workers returning home. Increased morning darkness in Permanent DST could exacerbate fatigue in shift workers [26,27]. Since darkness is known to contribute to crash risk [13,14,54], ensuring ambient light conditions during the morning rush hour would be the safest option to prevent risk at a time when there are not only daytime workers on the road but also fatigued shift workers returning home, student drivers, and buses full of school children. Work and school schedules could be modified to avoid dark morning commutes under any time change arrangements but are most closely aligned to this goal under CTA or permanent ST conditions.

This increase in darkness around the time of morning awakening is a strong argument against permanent DST [4–6]. In the absence of schedule constraints or artificial light, humans naturally awake around or after sunrise [18]. A mismatch between the timing of sleep due to schedule constraints and a human's natural circadian rhythmicity can result in recurrent symptoms of fatigue known as "social jet lag" [55,56]. Adolescents may be affected in particular due to a natural propensity towards later waketimes [57,58]. Early school start times have been known to disrupt student sleep and impair health and performance. Many states have introduced legislation to limit how early schools may start in the morning to curb this negative health effect [20,22,59]. Permanent DST would, in effect, undo the benefits of these efforts [60]. SAFTE-FAST takes potential daylight exposure, circadian misalignment, and sleep inertia into account to estimate Effectiveness, but the model has not been examined in the context of social jet lag. This constitutes a limitation for the current analyses and an interesting follow-up study to test in future investigations.

As expected, the effects of time change conditions on exposure to light differed by city location, season, and shift, as depicted in Figure 2 and shown in Supplementary Data Table S1. A limitation of this analysis is that we compare averages for Effectiveness and potential daylight exposure based on data from generic, hypothetical schedules and algorithmic predictions of sleep. This type of analysis cannot account for individual differences, rotating shift schedules, or behaviors specific to a certain population. The relationship between city selection criteria such as population, highway fatality rate, or distance relative to the start of the time zone on Effectiveness or sunlight could also not be examined in these analyses because the datasets are generic and hypothetical. The model-generated dataset also contains less variance than is expected from real-world data, which limits the interpretability of statistical significance. Differences between conditions were compared using the ANOVA *F* test, which is generally robust to violations of variance when sample sizes are equal [61], but the *p*-values should be examined in the context of group means, standard deviation, effect size, and ecological significance. Moreover, while longitude position relative to the time zone has recently been shown to impact traffic risk and social jet lag [62], the SAFTE-FAST model has not been developed or evaluated

for its ability to detect risk related to longitude position alone. An interesting follow-up study would be to evaluate the ability of SAFTE-FAST to model Effectiveness in cities with similar longitude positions but different time zones, such as Chicago and Indianapolis. In light of these limitations, it is important to note that if the Sunshine Protection Act is enacted, it will affect all people living in U.S. locations across the entire year regardless of their location, schedule, or individual differences. In this way, using hypothetical generic schedules may be a useful tool to evaluate the base level of risk associated with any time change arrangement.

Biomathematical models are frequently used in industry to prospectively investigate work schedules in order to avoid working during periods of high fatigue risk. Schedule adaptation has also been suggested for avoiding fatigue risk or circadian misalignment related to either permanent DST or ST [5,15]. Our findings suggest that the changes in light exposure or Effectiveness under permanent ST are more similar to CTA, particularly in student populations since school is traditionally not in attendance over the summer. Logistically speaking, permanent ST may require fewer schedule changes than DST and, therefore, make for an easier adjustment. An alternative interpretation is that neither permanent DST nor permanent ST offers a significant advantage over CTA. Adopting permanent ST would require fewer schedule changes than adopting permanent DST, but continuing to use CTA would require no schedule changes since the U.S. already uses this time change arrangement. According to a poll by The Associated Press-NORC Center for Public Affairs Research, only 25% of Americans support continuing the use of CTA [63]. Despite mixed evidence or a lack of direct evidence that adopting permanent time arrangements in either direction would improve traffic safety, energy use, daylight exposure, or health outcomes, Americans do not seem to prefer CTA. If American voters want to stop the bi-annual clock changes, the least disruptive permanent time option would appear to be permanent ST.

## 5. Conclusions

The effect of permanent DST on Effectiveness and potential daylight exposure relative to CTA or permanent ST were compared using a biomathematical model of fatigue (SAFTE-FAST). Controlling for U.S. location, season, and shift type, permanent DST would result in greater morning darkness than CTA or ST with only nominal differences in Effectiveness. Under permanent DST, morning rush hour would have a greater percentage of darkness that could increase risk during a time period of reduced commuter Effectiveness due to night shift workers returning home and greater traffic congestion, given the overlap between day workers and students commuting to school and night shift workers commuting home. Permanent DST would require greater adaptation to new schedules and result in less exposure to daylight in the morning and across the day. Permanent DST does not demonstrate any benefits with regard to traffic safety, fatigue risk, or light exposure for students, shift workers, or the general day-working public.

**Supplementary Materials:** The following supporting information can be downloaded at https://www.mdpi.com/article/10.3390/safety9030059/s1, Table S1: Percent Darkness During Total Waking Day and During Sleep by City, Season, and Shift by Time Change Condition; Table S2: Average Effectiveness During Work Day and Commutes by City, Season, and Shift by Time Change Condition; Table S3: Average Effectiveness and Percent Darkness During Morning Rush Hour by City, Season, and Time Change Condition; Table S4: Average Effectiveness and Percent Darkness During Evening Rush Hour by City, Season, and Time Change Condition.

**Author Contributions:** Conceptualization, J.K.D. and S.R.H., methodology, J.K.D., J.C. and S.R.H.; formal analysis, J.K.D., J.C. and S.R.H.; data curation, J.K.D. and J.C.; writing—original draft preparation, J.K.D.; writing—review and editing, J.K.D., J.C. and S.R.H.; supervision, S.R.H. All authors have read and agreed to the published version of the manuscript.

**Funding:** This research received no external funding.

**Institutional Review Board Statement:** Not applicable.

**Informed Consent Statement:** Not applicable.

**Data Availability Statement:** Data are available upon request.

**Conflicts of Interest:** The Institutes for Behavior Resources provides licensing of SAFTE-FAST. Authors J. K. Devine and J. Choynowski are affiliated with the Institutes for Behavior Resources but do not benefit financially or non-financially from licensing sales of SAFTE-FAST software. Author S. R. Hursh is the inventor of the SAFTE-FAST biomathematical model, and a fraction of his compensation is based on sales of the software.

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
