# Peer review of "Potential Effects of Permanent Daylight Savings Time on Daylight Exposure and Risk during Commute Times across United States Cities in 2023–2024 Using a Biomathematical Model of Fatigue"

_safety, 2023_

Round 1

Reviewer 1 Report

Dear Authors, 

I have carefully reviewed the manuscript titled " Potential Effects of Permanent Daylight Savings Time on Day-light Exposure and Risk During Commute Times Using a Biomathematical Model of Fatigue" and found it to be an interesting study with valuable insights. However, I would like to bring to your attention some issues that need to be addressed before considering it for publication.

· It would be better to revise the title according to the geographical scope of the study or the specific timeframe.

· The introduction should be revised to clearly state the research question or hypothesis to provide a better understanding of the study's focus. While it briefly mentions providing computational data on the impact of time change arrangements, it fails to clearly articulate the specific purpose or objective of the research.

· The introduction should be revised to include a comprehensive overview of both the potential benefits and drawbacks of permanent DST, supported by relevant research and evidence.

· In the method section, the description of the SAFTE-FAST model is limited and lacks clarity. It would be helpful to provide a more detailed explanation of the model, including its underlying assumptions, limitations, and previous applications in similar studies.

· In the statistical analysis, the specific statistical tests used in the analysis (e.g., the type of repeated measures ANOVA, assumptions checked) are not explicitly mentioned.

· The presentation of results could be enhanced by providing more detailed tables or figures to clearly summarize the findings.

· The discussion could benefit from a more detailed interpretation and discussion of the results. Specifically, discussing the implications of the findings in relation to existing literature or theories would strengthen the discussion section.

· The conclusions should be directly supported by the results presented.

· This study does not provide specific recommendations for future studies to address the identified limitations or further explore the topic.

Author Response

I have carefully reviewed the manuscript titled " Potential Effects of Permanent Daylight Savings Time on Day-light Exposure and Risk During Commute Times Using a Biomathematical Model of Fatigue" and found it to be an interesting study with valuable insights. However, I would like to bring to your attention some issues that need to be addressed before considering it for publication.

Response: The authors thank the reviewer for their time and helpful comments. We have addressed the issues as described below. 

Comment 1: It would be better to revise the title according to the geographical scope of the study or the specific timeframe.

Response: The title has been changed to reflect the geography and time period of the computational analysis.

Comment 2: The introduction should be revised to clearly state the research question or hypothesis to provide a better understanding of the study's focus. While it briefly mentions providing computational data on the impact of time change arrangements, it fails to clearly articulate the specific purpose or objective of the research.

Response: The introduction has been restructured to clearly state the purpose of the study and research question.

Comment 3: The introduction should be revised to include a comprehensive overview of both the potential benefits and drawbacks of permanent DST, supported by relevant research and evidence.

            Response: A summary regarding the potential benefits and drawbacks of permanent DST, supported by relevant research and evidence, has been included in the Introduction on Pages 1-2, Lines 42-56.

Comment 4: In the method section, the description of the SAFTE-FAST model is limited and lacks clarity. It would be helpful to provide a more detailed explanation of the model, including its underlying assumptions, limitations, and previous applications in similar studies.

            Response: This information was previously included in the Introduction and has been moved to the Methods section.

Comment 5: In the statistical analysis, the specific statistical tests used in the analysis (e.g., the type of repeated measures ANOVA, assumptions checked) are not explicitly mentioned.

            Response: The statistical analysis section has been updated to explicitly measure the type of repeated measures ANOVA (over time rather than crossover) and assumptions checked on Page 6, lines 279-282.

Comment 6: The presentation of results could be enhanced by providing more detailed tables or figures to clearly summarize the findings.

            Response: Detailed results have been included in Tables 1-4 as well as the supplementary materials. The authors did attempt to convey the information in figure format, but given the lack of variance in the generated dataset, the figures did not enhance the presentation of the results.

Comment 7: The discussion could benefit from a more detailed interpretation and discussion of the results. Specifically, discussing the implications of the findings in relation to existing literature or theories would strengthen the discussion section.

Response: The Discussion section has been rewritten to focus on the implications of the findings in relation to existing literature and theories.

Comment 8: The conclusions should be directly supported by the results presented.

            Response: The authors have done their best to draw conclusions in direct support of the results presented. Moreover, limitations to the generalizability of the findings or conclusions have been described in detail in the Discussion section. The Conclusions section has been modified to more directly support the findings.

Comment 9: This study does not provide specific recommendations for future studies to address the identified limitations or further explore the topic.

            Response: Recommendations for future studies in relation to the current study’s limitations are included in the Discussion section on Pages 14-15. Suggested follow-up studies include: 1) investigating the impact of social jet lag under time change conditions; 2) comparing fatigue risk between cities with similar longitude position but different time zones; and 3) using a biomathematical model of fatigue to assess base fatigue risk prior in enacting any time change arrangements by geography. 

Reviewer 2 Report

This paper evaluated the potential effects of permanent daylight savings time on fatigue risk or traffic congestion. The topic is meaningful and interesting. But several parts should be reviesd before it can be published.

1. Section 2.3, in the generation of work schedule data, why do the authors use three typical shifts for all cities? Do the six cities have different shifts since their geographical location is very different.

2. Table 1, how to explain the expected morning waketime during evening and night shift types?

3. Obviously,  permanent DST may increase the percent darkness during daytime. However, how does this relate to fatigure risk or traffic congestion? The authors should explain more about the relationship between darness and fatigure or congestion,

4. The authors can discuss more about the advantages and disadvantages of DST, ST, and CTA. Theses three time arrangements should be compared to identify if a certain place is more suitable for one of them.

Author Response

This paper evaluated the potential effects of permanent daylight savings time on fatigue risk or traffic congestion. The topic is meaningful and interesting. But several parts should be revised before it can be published.

Response: The authors thank the reviewer for their time and feedback. We have attempted to address the issues as described below.

Comment 1: Section 2.3, in the generation of work schedule data, why do the authors use three typical shifts for all cities? Do the six cities have different shifts since their geographical location is very different.

            Response: Three typical work shifts were chosen to represent the scope of work schedules across major cities. While there is no census data on the distribution of shift work by individual cities in the United States, the Bureau of Labor Statistics does report national prevalence of shift work. The latest report by the Bureau of Labor Statistics in 2019 reported that 16% of workers work a non-daytime schedule, of which 6% of workers worked evenings, and 4% worked nights. The remaining workers had a rotating shift, a split shift, an irregular schedule, or some other schedule. Without any evidence to support the role of geographical location on the prevalence of shift work, we must make the assumption that at least some portion of the population of these major U.S. cities can reasonably be expected to work a day, evening, or overnight shift. The Bureau of Labor Statistics findings on prevalence of shift work in the U.S. has been added to the Methods section on work schedules on Page 4, lines 198-199.

Comment 2: Table 1, how to explain the expected morning waketime during evening and night shift types?

            Response: The explanation of morning waketimes for evening and night shift schedules was previously included in the Statistical Analysis section. This information has been moved to the Methods section on SAFTE-FAST (Page 3, lines 117-120) and the Results section (Page 6, lines 297-299) to reduce confusion. In short, AutoSleep predicts split sleep schedules for evening and night workers; the first waketime occurring during any modeled day served as the expected morning waketime in these subgroups.

Comment 3: Obviously, permanent DST may increase the percent darkness during daytime. However, how does this relate to fatigue risk or traffic congestion? The authors should explain more about the relationship between darkness and fatigue or congestion.

Response: The impact of darkness on traffic accident risk is outlined in the Introduction section and addressed in the context of the results in the Discussion section. In brief, darkness is a known contributor to accident risk, and permanent DST conditions resulted in greater darkness during the mornings and overall waking day. The Discussion section has been rewritten to highlight the interesting finding that Effectiveness was predicted to be lower during morning rush hour compared to evening rush hour and number of commuters on the road was predicted to be higher due to shift workers returning home and students commuting to school. Time change arrangements did not impact Effectiveness during morning rush hour directly, but taken in tandem, these findings suggest that actions should be taken to minimize darkness during morning commutes to reduce accident risk.

Comment 4: The authors can discuss more about the advantages and disadvantages of DST, ST, and CTA. These three time arrangements should be compared to identify if a certain place is more suitable for one of them.

            Response: The reviewer has an excellent point that geography may influence the appropriateness of time change arrangements. Our analysis did not indicate a location effect, and importantly, the Sunshine Protection Act also does not take location into account. These points have been addressed throughout the Discussion section.

Reviewer 3 Report

The idea of the paper seems interesting and timely; however, I am not sure about the scientific contribution to the body of knowledge, given that the research is based on utilizing an available software application (SAFTE-FAST) to generate data based on generic inputs. This may make an interesting case study, but I am not sure if it can make a full-length paper. Additionally, I am providing the following comments to help the authors improve the paper:

1.       The introduction section is too lengthy and needs to be better organized. The last paragraph of the introduction section (about the Sunshine Protection Act) was already discussed at the beginning of the introduction.   

2.       There is an incorrect reference to Section 4.2 on page 4 (line 192).

3.       I think the reference to Section 2.3 on page 4 (line 199) was meant to refer to Section 2.2.

4.       The results and the conclusions do not really add any more insights than what can be obtained by a layman using simple observations without the need to use the SAFTE-FAST software. Again, this is a question about the scientific contribution of the paper. 

Some minor language improvements are needed 

Author Response

The idea of the paper seems interesting and timely; however, I am not sure about the scientific contribution to the body of knowledge, given that the research is based on utilizing an available software application (SAFTE-FAST) to generate data based on generic inputs. This may make an interesting case study, but I am not sure if it can make a full-length paper. Additionally, I am providing the following comments to help the authors improve the paper:

Response: The authors thank the reviewer for their time and feedback. We have attempted to describe in more clear terms the importance of demonstrating objective differences in daylight exposure and fatigue risk under differing time change arrangements not only for the scientific community but also for the political community who rely on scientific documentation to support decision-making. We have done our best to address the reviewer's specific comments below.

Comment 1: The introduction section is too lengthy and needs to be better organized. The last paragraph of the introduction section (about the Sunshine Protection Act) was already discussed at the beginning of the introduction.   

Response: The Introduction section has been organized to be more concise and clearly worded.

Comment 2:  There is an incorrect reference to Section 4.2 on page 4 (line 192). I think the reference to Section 2.3 on page 4 (line 199) was meant to refer to Section 2.2.

            Response: These references has been corrected.

Comment 3:  The results and the conclusions do not really add any more insights than what can be obtained by a layman using simple observations without the need to use the SAFTE-FAST software. Again, this is a question about the scientific contribution of the paper.

            Response: This analysis is the first to use biomathematical modeling to examine the potential impact of time change conditions on fatigue risk and light exposure during future timepoints across multiple locations and shift working or student conditions. The results and conclusions are very much in line with the scientific community’s consensus opinion about the impact of permanent DST, but has the benefit of providing objective computational data to support those insights. The purpose of this analysis has been to fill a research gap within the argument for or against the adoption of permanent DST under the Sunshine Protection Act—namely, what is the expected difference in daylight exposure and fatigue risk based solely on time change arrangements without the interference of confounding variables such as chronic sleep loss, seasonal variability, or individual differences. This paper moreover outlines a potential tool—biomathematical modeling-- that can be used to evaluate schedules under these conditions in the future. The authors feel that these attributes of the paper justify its scientific contribution to the literature.

Round 2

Reviewer 2 Report

I think the authors have already revised the paper properly.

Reviewer 3 Report

My comments have been addressed

Good quality